# A dense MEMS-based seismic network in populated areas: rapid estimation of exposure maps in Trentino (NE Italy)

Davide Scafidi[1], Alfio Viganò[2], Jacopo Boaga[3], Valeria Cascone[3], Simone Barani[1], Daniele Spallarossa[1], Gabriele Ferretti[1], Mauro Carli[4], Giancarlo De Marchi[4]

[1]Dipartimento di Scienze della Terra, dell'Ambiente e della Vita, University of Genoa, Corso Europa 26, 16132 Genoa, Italy

[2]Servizio Geologico, Provincia autonoma di Trento, Via Zambra 42, 38121 Trento, Italy

[3]Dipartimento di Geoscienze, University of Padova, Via Gradenigo 6, 35131 Padova, Italy

[4]AD.EL. s.r.l., via Sandro Pertini 5, 30030 Martellago (VE), Italy

*Correspondence to*: Davide Scafidi (davide.scafidi@unige.it)

**Abstract**

The MEMS-based seismic network of Trentino (NE Italy) consists of 73 low-cost accelerometers installed close to inhabited areas. These sensors have a suitable sensitivity to detect moderate-to-strong earthquakes but are able to record even weaker seismicity. The densely distributed peak ground acceleration values recorded by MEMS and other types of stations are integrated within the existing seismic monitoring procedure in order to automatically obtain a complete set of strong motion parameters a few minutes after the origin time. The exposure for resident population and critical buildings is estimated by quantifying the different levels of shaking, which is expressed according to the Mercalli-Cancani-Sieberg intensity scale. These types of results, summarized in synthetic PDF (Portable Document Format) documents, can be useful for civil protection purposes to timely evaluate the state of emergency after a strong earthquake and to choose how and where activate first aid measures and targeted structural monitoring.

## 1 Introduction

During the last decades seismic monitoring has been greatly improved in order to give precise and increasingly detailed information for emergency and environmental purposes. Besides permanent seismic networks, a primary role in capturing the increased amount of instrumental data is given by low-cost micro-electromechanical system (MEMS) instrumentation (D'Alessandro et al., 2019). Nowadays, MEMS accelerometers are widely used on different spatial scales to replace or densify permanent networks, in order to improve seismic detection and evaluate with greater resolution the effects of earthquakes (Cochran et al., 2009; Boaga et al., 2018; Patanè et al., 2022; Vitale et al., 2022). Earthquake early warning systems have also been benefitting greatly from MEMS technology, because targeted timely actions can be automatically taken in case of strong earthquakes (Satriano et al., 2011; Cochran, 2018). For this reason, large earthquake datasets need to be efficiently and rapidly managed (Spallarossa et al., 2021) and related outcomes (e.g., earthquake location and magnitude, strong motion data and maps) shared in real-time with different end users, such as scientists, technicians, politicians, civil protection, decision makers, and citizens.

The Trentino region (NE Italy) is currently monitored by a permanent seismic network, which has been managed by the Autonomous Province of Trento (PAT) since 1981 (Geological Survey–Provincia Autonoma di Trento, 1981; Viganò et al., 2021; Fig. 1). According to the Italian building code (Ministero delle Infrastrutture e dei Trasporti, 2018) this area is characterized by peak ground acceleration (PGA) values lower than 0.18 g (for a return period of 475 years), with highest seismic hazard in southern Trentino (upper Lake Garda and lower Adige Valley) and eastern Trentino (lower Valsugana, Tesino and Primiero) where fault systems are mostly active (Viganò et al., 2015) (Fig. 2). The resident population on 1[st] January 2022 is 540,958 (ISTAT, 2012) and is mostly concentrated in the city of Trento and along the main valleys where principal road networks and infrastructures are located.

Here, we present a local network based on MEMS accelerometers in Trentino, aimed at real-time monitoring and automatic generation of exposure maps. Co-seismic recordings are automatically processed and integrated with those from other stations (e.g., belonging to other permanent networks), allowing for a dense distribution of ground motion measurements.

## 2 Method

Maps displaying seismic shaking are widely used during emergency due to their ability to summarize earthquake effects and their potential impact on local targets (Michelini et al., 2020). In order to lead effective emergency actions, it is essential that these maps, named "exposure maps" hereafter, are available in a few minutes after a seismic event. In fact, they provide a first-level overview of the expected damage over the monitored area.

The exposure maps of the Trentino civil protection are automatically generated by using all the available seismic data (i.e., ground motion measurements), with the aim of estimating the asset exposed to an earthquake (Fig. 3). In particular, MEMS recordings are integrated with those from other stations and used to obtain a complete set of strong motion data, in order to quantify the numbers of resident population and buildings subjected to different levels of shaking. A step-by-step description of the method used to generate the exposure maps is given in the next sections.

58

## 2.1 MEMS accelerometer design and installation

The low-cost MEMS sensor adopted in the presented network is the ADXL355 of the Analog Device. AD.EL s.r.l., an Italian based telecommunication company, developed the board for housing and operating the MEMS accelerometer, named "ASX1000v2" (D600158 AD.EL code; Fig. 4a). The ASX1000v2 is a capacitive triaxial accelerometer, conceived to be a platform for data acquisition and recording for long-term measurements. It is equipped with a high-performance MicroController Unit (MCU; STM32H743 model by STMicroelectronics) and communication channels for remote control and data transmission: a serial channel RS-422 or RS485, a LAN Ethernet 10/100 Mbit/s, an USB 2.0, and a 4G LTE modem (Fig. 4b). This sensor operates in high sensitivity mode for an acceleration range of ±2 g (it supports also the ±4 g full scale configuration), with a 250 Hz sampling rate. Time synchronization is obtained using the Network Time Protocol (NTP). Data streams from each single station are collected by a dedicated server; here, data are formatted, stored and made available for the automatic processing by using a standard SeedLink server.

The noise analysis relative to each component reveals a Power Spectral Density with a general downward trend between −80 and −65 dB in the 0.03–10 Hz frequency range (Fig. 5). As shown in Figure 5, the detectability threshold of seismic events corresponds to a moment magnitude of about 3.5. Therefore, this sensor has a suitable sensitivity to detect moderate-to-strong events, those that are of primary interest to public administration for emergency management.

The MEMS sensors are installed inside telecommunication infrastructures. Each sensor is firmly coupled with the ground with screws and plugs, at the base of the local server room; the azimuth is carefully measured during installation. Each sensor is plugged into a wall outlet for power. A complete station costs only a few hundred euros, making possible the deployment of dense arrays of accelerometers.

## 2.2 Data integration and seismic processing

Seismic data processing is here performed by using the software CASP – Complete Automatic Seismic Processor (Scafidi et al., 2016; 2018; 2019). By taking advantage of the features of its iterative procedure, this software can effectively manage (during phase picking and location) data provided by different seismic stations with variable signal quality. Contrary to stations of permanent monitoring networks, which are usually installed in remote and quiet areas to ensure seismic signals with low noise levels, signals from seismic stations deployed in urban areas, such as those from our MEMS network, can be significantly affected by high level noise (producing spikes and impulsive signals) due to anthropogenic activities. This may lead to an uncontrolled proliferation of false (i.e., non-seismic triggers). Therefore, their use in automatic phase picking procedures may affect the reliability of the final earthquake location and, in some cases, lead to false events. Hence, noisy stations are often neglected in automatic earthquake monitoring. CASP processes signals by using an iterative procedure within which the phase picking is driven by earthquake location (Spallarossa et al., 2014). On the one hand, this allows identification of false triggers. On the other hand, arrival times are improved at each iteration, leading to an optimization of the earthquake location.

With reference to the present application, which integrates data from permanent monitoring networks and data from the
MEMS stations, CASP is set not to use MEMS data in the first iteration of the location procedure, thus assuming that they
are affected by significant background noise. In this step, the definition of arrival times is not yet driven by location but it is
based on an envelope function on signals (Spallarossa et al., 2014). This precaution may not be necessary for local strong
earthquakes, for which the seismic signal clearly dominates the background noise, but it is useful when managing signals
from weak earthquakes. From the second iteration on, signals from all stations are used and P- and S-wave arrivals are
computed by applying the Akaike Information Criterion – AIC (Akaike, 1974) on signal windows centred, for each station,
around the expected arrival times obtained by the location code. In fact, these picks are determined (at each iteration) by the
location algorithm working in conjunction with CASP, the NonLinLoc software (Lomax et al., 2000). This allows to reliably
discriminate between seismic phase arrivals and signal disturbances also in the case of weak-to-moderate earthquakes
recorded by different stations, regardless of the type of sensor used.
In addition to the computation of hypocentral parameters, for each station with at least one phase picked, CASP returns the
values of a number of ground motion parameters (e.g., PGA, peak ground velocity PGV, spectral acceleration).
In the case of the Trentino region, a fully automated earthquake monitoring has been already operating based on CASP
(Viganò et al., 2021). Thus, the great amount of data provided by the 73 installed MEMS stations (starting date July 2022;
Fig. 1) has been easily integrated within the seismic monitoring procedure as the only requirements for CASP are real-time
data transmission in standard SeedLink format and station response metadata in seismological standard format (i.e., Dataless,
StationXML, Poles and Zeroes – PAZ file). About data transmission between the MEMS stations and the central processing
system, the typical average latency is in the order of about 15 s, while the data stream of all the MEMS stations is continuous
and complete at about 99.5 %.

## 2.3 Exposure maps
Exposure maps are automatically created using the GMT software (Wessel and Smith, 1998) and the PHP open-source
scripting language. At first, shaking data recorded by each station (i.e., peak ground accelerations) are converted to intensity
values (Mercalli-Cancani-Sieberg scale, MCS) using empirical relationships for Italy (Faenza and Michelini, 2010 for PGA
<1 cm s$^{-2}$; Oliveti et al., 2022 for PGA $\geq$1 cm s$^{-2}$). Intensity, which is considered more informative than peak ground
acceleration for civil protection purposes as it is directly based on earthquake damage and perception, is colour coded
according to the ShakeMap palette (Michelini et al., 2020). These densely distributed data are then gridded using adjustable
tension continuous curvature splines ("surface" routine command in GMT, with tension set to 0.5), with no pre-processing
(e.g., blockmean) or interpolation. This is possible because of the dense distribution of MEMS stations, which are mainly
located in the vicinity of inhabited areas. At this stage, a maximum intensity value is assigned to each municipality in
Trentino, for which the cumulative number of resident population is known (Fig. 6). Then, the intensity map is compared to
the distribution and density of resident population in Trentino (last national census; ISTAT, 2012), where territorial localities
are classified as (i) urban area, (ii) small inhabited areas, (iii) productive areas or (iv) wide spread houses. For each locality

the procedure automatically calculates the maximum intensity and combines it with the population density. The cumulative

population for each intensity level is then computed. In a similar way, the system automatically processes (as polygonal

features) the distribution of buildings of interest for the Autonomous Province of Trento (Fig. 6), and the cumulative number

of buildings for each intensity class is obtained. Finally, peak ground acceleration is measured at 16 instrumented dams

located in Trentino (Fig. 6). As the strong motion parameters from all the other stations, also these ones are converted to

intensity values and used to create the Trentino exposure maps.

**3 Results**

The estimation of exposure maps in Trentino is usually carried out within 10 minutes from an earthquake. A local magnitude

($M_L$) threshold for their automatic generation is set to $M_L$ 4.0. The procedure has been activated since July 2022, using a

standard workstation equipped with an Intel Core i5 CPU. Even if no strong earthquakes occurred until now (October 2023)

in the monitored area, MEMS stations have been used for standard locations (i.e., available additional phase arrivals from

MEMS stations are used by the location procedure) and to record the ground motions of low-to-medium energy seismic

events. We note that a seismic signal recorded by a MEMS station is commonly clearly detectable for events with $M_L$ greater

than about 2.5, considering hypocentral distances of a few tens of kilometres (compare also with results by Cascone et al.,

2021). In fact, even if the MEMS application presented in this study is principally aimed to perform quasi real-time exposure

maps in the urbanized areas of Trentino, in Appendix A the low magnitude earthquakes which were recorded by at least one

MEMS station during the period July 2022–October 2023 is listed. In some cases, some stations recorded a readable signal,

related both to seismic events inside or outside the Trentino area. As an example, we can consider the automatically detected

P- and S-phase arrival times (red and blue vertical lines in Fig. 7, respectively) for the $M_L$ 2.7 earthquake occurred on

November 10[th] 2022 in the Fassa Valley (NE Trentino). GAGG is a standard seismic station of the permanent PAT network,

while station 003B belongs to the MEMS network (see Fig. 1). Both stations are located in the same area (2 km apart from

each other) at about 65 km from the earthquake hypocentre. Even if the P-phase onset for station 003B is masked by the

background noise, which is clearly higher than the noise affecting the GAGG recordings, the CASP procedure is able to

detect the S-phase arrival time. Thus, both GAGG and 003B can be used to calculate the strong motion parameters for that

event (Fig. 8). Few minutes (maximum 5) after the origin time, CASP returns event location, magnitude, and the strong

motion table (for all the analysed stations), which includes: PGA, PGV, Peak Ground Displacement (PGD), Spectral

Acceleration (SA) for different response periods (T), response spectrum intensity (also known as Housner Intensity, IH) for

different period ranges (0.1–0.5 s, IH 0; 0.1–1.0 s, IH 1; 0.1–1.5 s, IH 2), and Instrumental Intensity ($I_{MCS}$; Mercalli-Cancani-

Sieberg scale). Compared to station GAGG, station 003B shows stronger shaking values that can be attributed to the effect

of different subsoils (Fig. 8). As with all stations belonging to the PAT permanent network, GAGG is deployed on bedrock,

while 003B is located in the middle of an alluvial valley near the town of Vezzano. Here, alluvial deposits are reasonably

assumed to be responsible of the observed shaking amplification. The higher ground motion values of station 003B are used

for a site-specific exposure map, which can take into account local seismic effects near towns and populated areas.

The exposure maps and all the relevant seismic results provided by CASP are reported in an automatically generated document in standard PDF (Portable Document Format) format, which also contains links to the high resolutions maps stored online. This summary file represents an easy and user-friendly mean of communications that can be easily disseminated through emails and messaging platforms (e.g., Telegram), read online, or printed. Figure 9 shows the PDF of the exposure map generated for an $M_L$ 2.1 earthquake occurred on July 11[th] 2023 in Western Trentino. After a synthetic textual and graphical summary of event location (magnitude, area, origin time and hypocentral data), tables and maps relative to the seismic shaking and exposure are displayed. The first table contains a quantification of the population and the number of buildings of interest (A and B levels according to the administrative classification) possibly stricken by the earthquake for each intensity level. The maximum recorded intensity is VI MCS at about 5 km from the earthquake hypocentre (which is only 4.8 km deep). Of note, without the information provided by the MEMS network, we would have significantly underestimated the maximum intensity induced by the earthquake, which would not have exceeded III MCS. The PDF also shows two intensity maps that can be helpful for a rapid inspection of the damaged area. The first one shows interpolated values while the second one displays the values actually observed at each analysed station. Besides the maps, two tables provide further details about the measured shaking levels for both potentially involved population (first 20 municipalities sorted according to decreasing intensity) and available instrumented dams (listed according to both decreasing intensity and PGA values).

In order to test the procedure considering a realistic emergency scenario for a moderate event, we have simulated an $M_L$ 5.8 earthquake in Southern Trentino (45.834 °N latitude, 11.066 °E longitude, 9.0 km depth). This event has been selected to roughly simulate the so-called "Middle Adige Valley" earthquake, which represents a reference for the seismic potential of the Trentino region, as also evidenced by recent studies (e.g., Ivy-Ochs et al. 2017 and references therein). This earthquake dated to 1046 AD, with estimated epicentral intensity IX MCS and co-seismic shaking responsible for great damage and catastrophic induced events. The performed calculation represents a simplified simulation, obtained by assigning the selected event magnitude and then calculating PGA at each seismic station of the network (MEMS and permanent stations). PGA is computed using the regional attenuation law developed within the framework of the INGV-DPC Project S4 (Michelini et al., 2008). In particular, the regionalized attenuation relation adopted for the Eastern Alps is used. The summary PDF document relative to this earthquake is shown in Figure 10. According to this scenario (possibly even worse than presented, because of the simplified approach used), about 60 thousand people and 262 buildings of interest are involved in the area with maximum intensity (VIII MCS); the four municipalities with maximum intensity count a total population of about 52,000 people. Concerning dams, two of them reach PGA values greater than 0.3 g; this is important in order to define specific structural monitoring when predetermined PGA thresholds are overcome.

**4 Summary and conclusions**

We have presented an upgrade of the seismic monitoring procedure of the Trentino region through the integration of data provided by 73 low-cost MEMS accelerometers installed in urban areas. This dense MEMS-based network has a suitable

sensitivity to detect moderate-to-strong seismic events; weaker earthquakes with local magnitude lower than 3.0 can be even recorded and analysed. The additional data in conjunction with the automatic monitoring procedure currently in use allows us to obtain a densely distributed set of strong motion measurements and, consequently, high-definition shaking maps that relies only on actual recorded data. Integrating these dense MEMS data, though noisy, allows avoiding the use of ground motion prediction equations, thus leading to a more reliable picture of the actual ground shaking (hence, of the expected damage). This is of paramount importance for post-earthquake emergency planning in densely populated, urbanized areas characterized by high seismic risk. The use of the CASP code is crucial to properly manage such noisy data with the aim of getting reliable results in quasi real-time.

In addition to shaking data, the procedure presented here provide automatically generated exposure maps that quantify the resident population and the number of critical buildings in Trentino, subjected to different levels of shaking during an earthquake. Exposure maps are reported in synthetic PDF documents, which are very useful for civil protection in order to rapidly evaluate the local state of emergency after a strong earthquake and to choose how and where activate first aid measures, both for population and buildings of interests like dams.

## Code availability

The Complete Automatic Seismic Processor (CASP) is a commercial software.

## Author contributions

DS, AV, JB and MC conceptualized the project; JB, VC, MC and GDM developed the MEMS sensor; DS, AV, JB and MC followed MEMS installation; DS, AV, MC and GDM performed data integration; DS and GF made the earthquake simulation; DS, AV, JB, GF and SB wrote the manuscript draft; DS, AV, JB, GF, SB and DSp edited the manuscript; DS, AV, VC and GDM revised the manuscript.

## Competing interests

The authors declare that they have no conflict of interest.

## Acknowledgements

The authors gratefully acknowledge Domenico Patanè and two anonymous reviewers for their helpful comments and suggestions. This research was supported by the Geological Survey of the Autonomous Province of Trento (www.protezionecivile.tn.it). TIM (Telecom Italia Mobile) is gratefully acknowledged for supporting AD.EL during installation of MEMS stations. Maps were made using Generic Mapping Tools v.4.5 (Wessel and Smith, 1998).

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

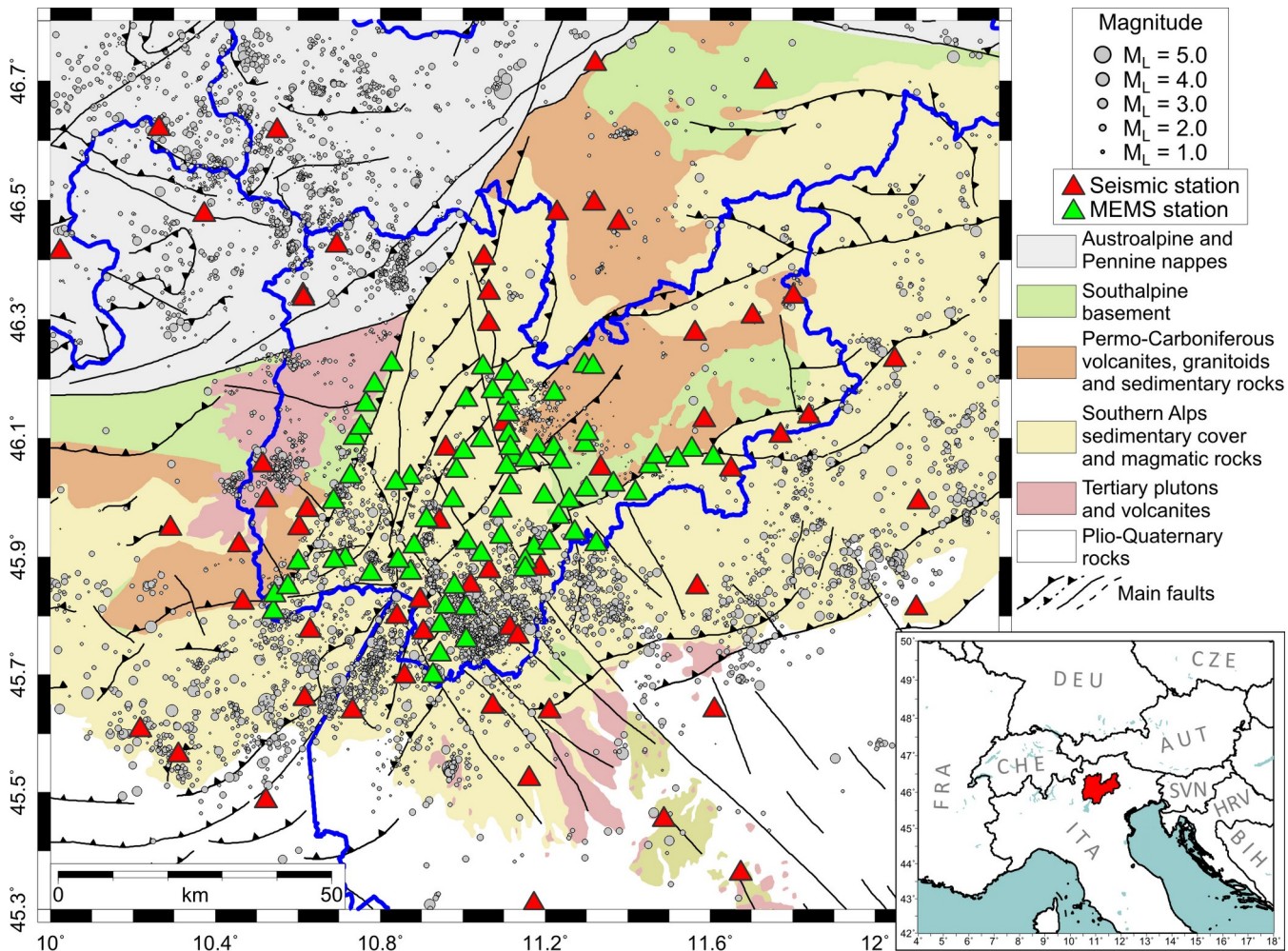

Figure 1: Simplified geological map of the Trentino region with epicentral distribution of earthquakes in the period 1981-2021 and local seismic networks. Green triangles represent the MEMS-based network (73 stations at October 2023).

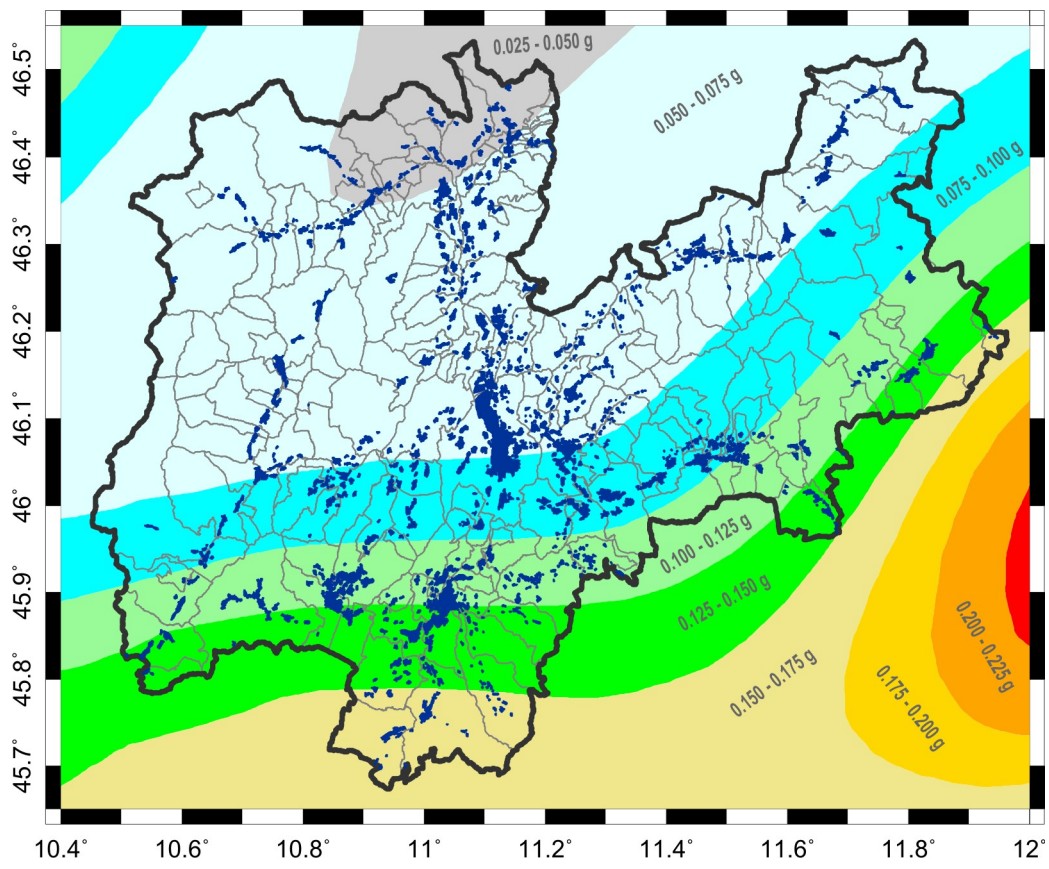

**Figure 2: Seismic hazard map showing the peak ground acceleration for a return period of 475 years (10%**
**probability of exceedance in 50 years) (Stucchi et al., 2011). Localities highlighted in dark blue (ISTAT, 2012).**

| Collection of seismic data from different networks (real-time) | Automatic elaboration with "CASP" software (quasi real-time) | Exposure maps creation and dissemination (straight gridded interpolation of measured data; quasi real-time) |

**Figure 3: Flowchart showing the process behind the generation of the exposure maps for the Trentino region.**

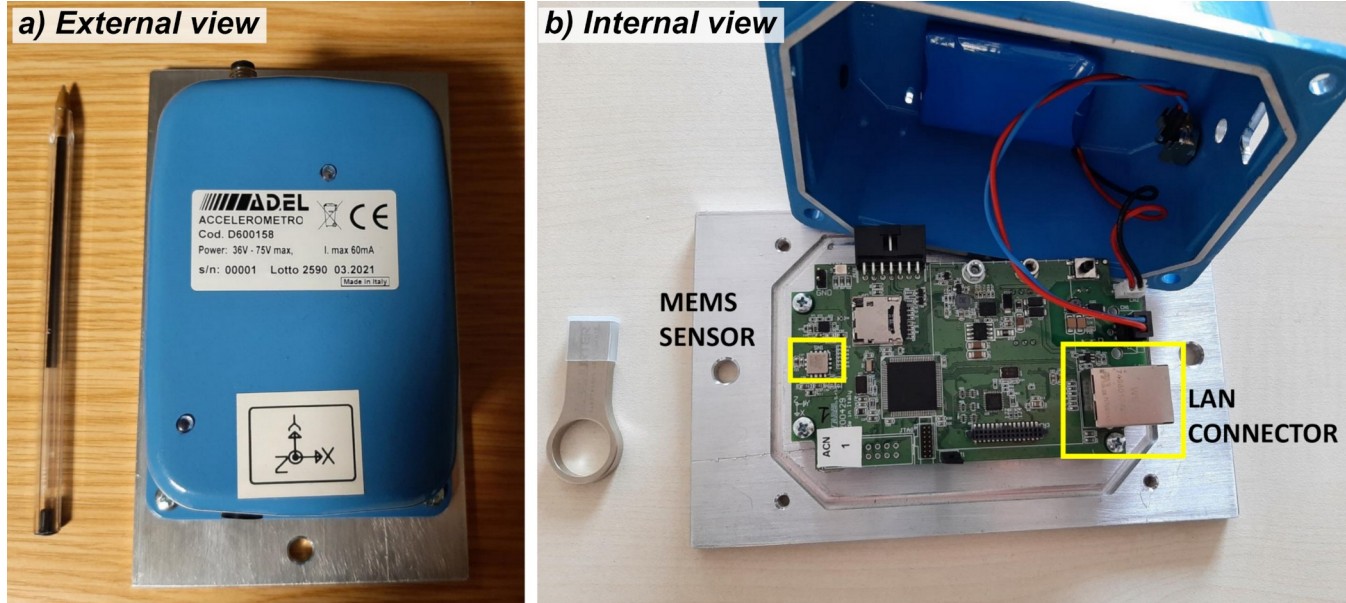

Figure 4: (a) The ASX1000v2 MEMS sensor prototype; (b) internal circuit batch.

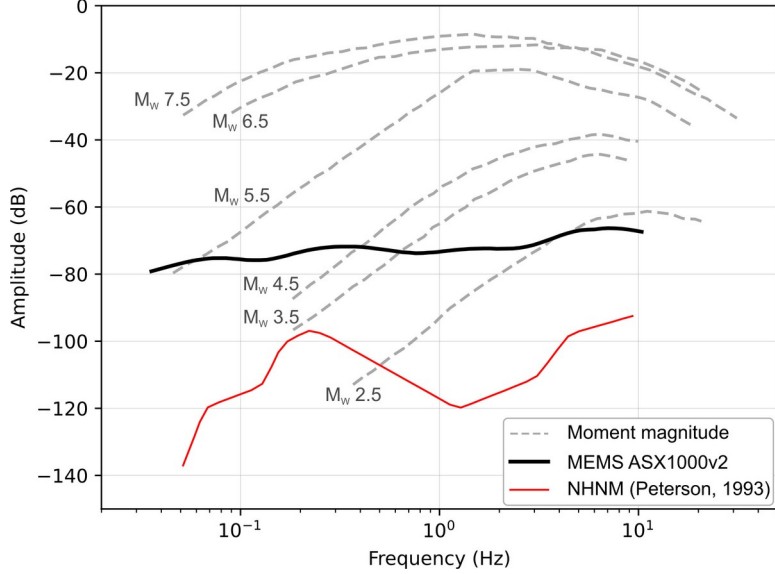

Figure 5: Noise floor of the ASX1000v2 MEMS (black line) compared to typical ground motion amplitudes of earthquakes measured at 10 km from the epicentre for different moment magnitudes (dashed lines). The new high noise model (NHNM – red line) from Peterson (1993) is also shown for reference.

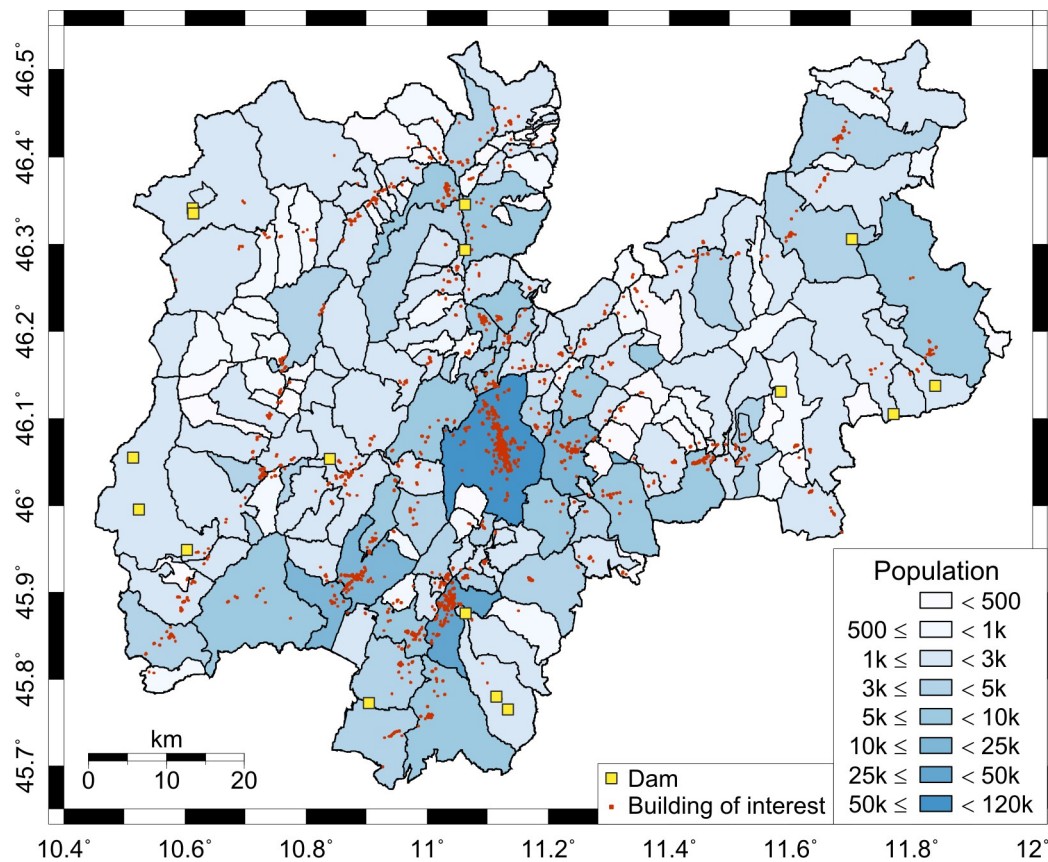

**Figure 6: Trentino municipalities coloured according to the resident population density (ISTAT, 2012), with buildings of interest (red dots) and main dams (yellow boxes) highlighted.**

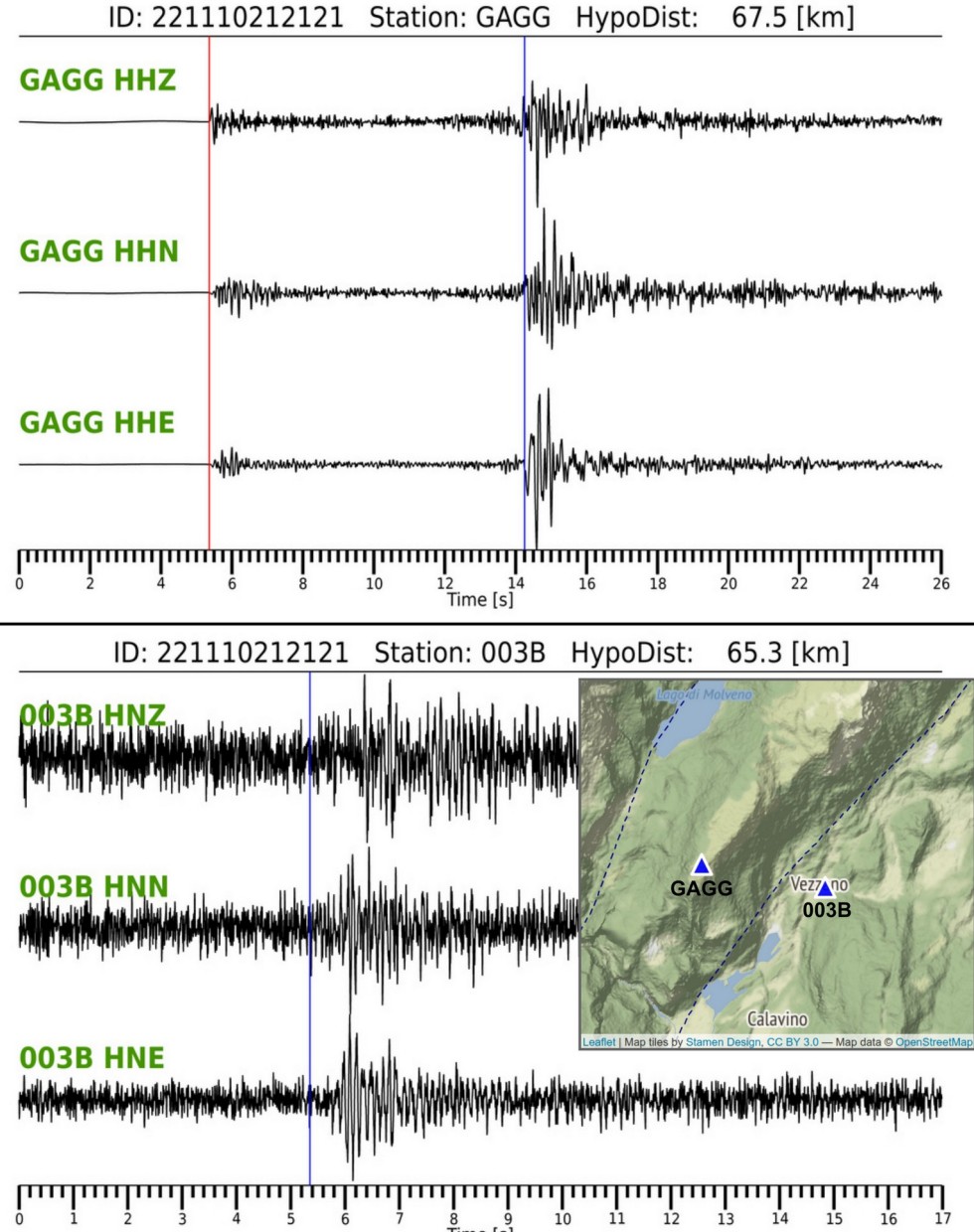

**Figure 7: Unfiltered three-component seismic traces from standard (GAGG) and MEMS sensors (003B) (see their geographic location in the inset) associated with automatically detected P- and S-phase arrival times (red and blue lines, respectively).**

| Station | Net | Chan. | PGA (g) | PGV (m/s) | PGD (m) | IH 0* (m) | IH 1* (m) | IH 2* (m) | Sa(T=0.10) (g) | Sa(T=0.30) (g) | Sa(T=1.00) (g) | Sa(T=3.00) (g) | Dist. (km) | Azim. (°) | $I_{MCS}$ |
|---|---|---|---|---|---|---|---|---|---|---|---|---|---|---|---|
| GAGG | ST | HNZ | 1.1544e-4 | 2.2126e-5 | 6.2110e-7 | 1.0179e-5 | 1.9381e-5 | 2.8232e-5 | 2.1453e-4 | 3.5065e-5 | 2.1356e-6 | 4.2307e-7 | 67.4 | 234 | - |
| GAGG | ST | HNN | 2.9669e-4 | 6.0573e-5 | 2.2308e-6 | 4.2506e-5 | 6.9291e-5 | 6.9291e-5 | 5.6295e-4 | 1.3750e-4 | 8.1487e-6 | 1.3107e-6 | 67.4 | 234 | - |
| GAGG | ST | HNE | 1.6050e-4 | 3.0075e-5 | 9.4923e-7 | 2.0460e-5 | 3.6440e-5 | 5.1543e-5 | 3.2503e-4 | 7.9317e-5 | 4.1464e-6 | 7.4897e-7 | 67.4 | 234 | - |
| GAGG | ST | HHZ | 6.2145e-5 | 1.2363e-5 | 3.1840e-7 | 5.5445e-6 | 1.0630e-5 | 1.5508e-5 | 1.1933e-4 | 1.8409e-5 | 1.1905e-6 | 2.3914e-7 | 67.4 | 234 | - |
| GAGG | ST | HHN | 1.8374e-4 | 3.3499e-5 | 1.0534e-6 | 2.2252e-5 | 3.9742e-5 | 3.9742e-5 | 3.6630e-4 | 8.2173e-5 | 4.3522e-6 | 8.3149e-7 | 67.4 | 234 | - |
| GAGG | ST | HHE | 3.4416e-4 | 6.5648e-5 | 2.4004e-6 | 4.6288e-5 | 7.5544e-5 | 1.0331e-4 | 6.5411e-4 | 1.4711e-4 | 8.9149e-6 | 1.4392e-6 | 67.4 | 234 | - |
| 003B | TN | HNZ | 7.3837e-4 | 1.7043e-4 | 1.5931e-5 | 1.1293e-4 | 2.6258e-4 | 3.6550e-4 | 2.9278e-3 | 4.2887e-4 | 1.2247e-4 | 8.3564e-6 | 65.2 | 232 | 1.3 |
| 003B | TN | HNN | 6.3724e-4 | 9.0012e-5 | 9.0852e-6 | 6.8410e-5 | 1.4765e-4 | 1.4765e-4 | 2.3232e-3 | 2.9415e-4 | 6.2978e-5 | 7.1631e-6 | 65.2 | 232 | 1.2 |
| 003B | TN | HNE | 9.8603e-4 | 1.6420e-4 | 7.6455e-6 | 1.0939e-4 | 2.0714e-4 | 2.9276e-4 | 4.5366e-3 | 2.8501e-4 | 1.1568e-4 | 7.3048e-6 | 65.2 | 232 | 1.6 |

**Figure 8: Screenshot of the automatically created summary table with strong motion data from standard (GAGG) and MEMS sensors (003B). Net, network; Chan., recording channel; Dist., hypocentral distance; Azim., azimuth; see text (section 3) for the other parameter abbreviations and meaning.**

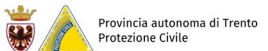 Provincia autonoma di Trento
Protezione Civile

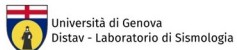 Università di Genova
Distav - Laboratorio di Sismologia

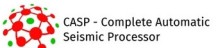 CASP - Complete Automatic
Seismic Processor

MAGNITUDE (M_L): 2.1
Area: Trentino_SW_Lago_di_Garda_e_Lessini
Origin Time: 2023/07/11 14:20:00 (GMT +0)
Epicentre: 46.027 (°N) ; 10.738 (°E)
Depth: 4.8 (km)

EPICENTRE

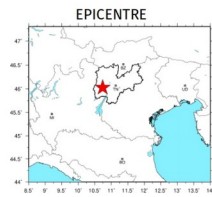

### Seismic shaking exposure

| Intensity (I_MCS): | ≤ III | IV | V | VI | VII | VIII | IX | X | ≥ XI | |
|---|---|---|---|---|---|---|---|---|---|---|
| Perceived Shaking: | Very light | Light | Moderate | Quite strong | Strong | Very strong | Severe | Very severe | Extreme | |
| Population[1]: | - | 1.8K | 3.7K | 0 | 0 | 0 | 0 | 0 | 0 | Total: 5.6k |
| Buildings of interest A[2]: | - | 9 | 14 | 0 | 0 | 0 | 0 | 0 | 0 | Total: 23 |
| Buildings of interest B[2]: | - | 2 | 21 | 0 | 0 | 0 | 0 | 0 | 0 | Total: 23 |

[1]ISTAT 2011 census estimation; [2]PAT 2022 census estimation.

### SHAKING INTENSITY (interpolated values)

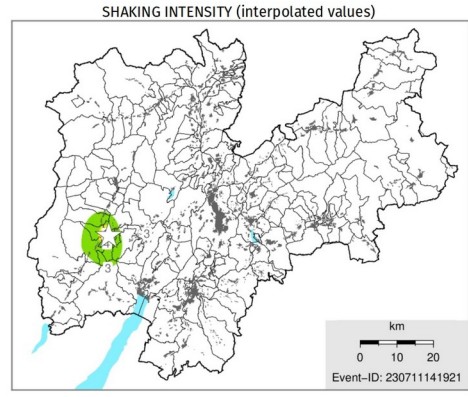

Event–ID: 230711141921

### MUNICIPALITIES EXPOSURE (first 20)

| I_MCS | Municipality | Population |
|---|---|---|
| V (5.4) | TIONE DI TRENTO | 3.665 |
| IV (4.8) | BORGO LARES | 707 |
| IV (4.7) | TRE VILLE | 1.404 |
| IV (4.2) | SELLA GIUDICARIE | 2.894 |
| III (3.9) | PORTE DI RENDENA | 1.752 |
| III (3.5) | BLEGGIO SUPERIORE | 1.516 |
| III (3.2) | LEDRO | 5.248 |
| III (3.1) | PIEVE DI BONO-PREZZO | 1.430 |
| III (3.1) | FIAVÈ | 1.055 |
| III (3.1) | PELUGO | 390 |
| III (3.0) | COMANO TERME | 2.895 |
| III (3.0) | DRO | 5.057 |
| III (3.0) | SPIAZZO | 1.244 |
| III (3.0) | TENNO | 1.992 |
| III (3.0) | VALDAONE | 1.141 |
| < III (2.8) | RIVA DEL GARDA | 17.646 |
| < III (2.7) | STENICO | 1.178 |
| < III (2.4) | ARCO | 17.798 |
| < III (2.4) | BOCENAGO | 396 |
| < III (2.4) | STREMBO | 609 |

### SHAKING INTENSITY (actual values)

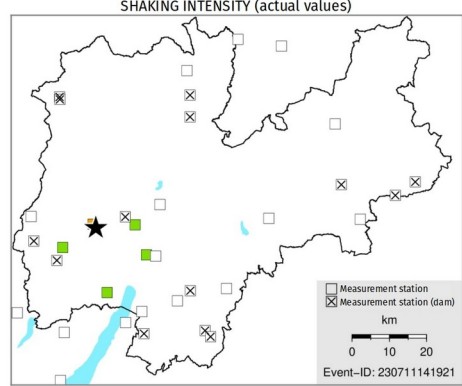

Measurement station
Measurement station (dam)

Event–ID: 230711141921

### DAMS (decreasing exposure)

| I_MCS | Accel. max (g) | Dams |
|---|---|---|
| < III (1.6) | 0.0009 | Malga Boazzo |
| < III (1.3) | 0.0007 | Ponte Pià |
| < III (1.2) | 0.0006 | Murandin |
| < III (0.0) | 0.0002 | Malga Giumela |
| < III (0.0) | 0.0001 | Pian Palù |
| < III (0.0) | 0.0001 | Mollaro |
| < III (0.0) | 0.0000 | Santa Giustina |
| < III (0.0) | 0.0000 | San Colombano |
| < III (0.0) | 0.0000 | Busa |
| < III (0.0) | 0.0000 | Pra da Stua |
| < III (0.0) | 0.0000 | Costabrunella |
| < III (0.0) | 0.0000 | Val Noana |
| < III (0.0) | 0.0000 | Speccheri |
| < III (0.0) | 0.0000 | Val Schener |

**Figure 9: Exposure map PDF for a weak earthquake occurred in Western Trentino. See text (section 3) for description.**

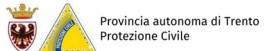
**Provincia autonoma di Trento**
Protezione Civile

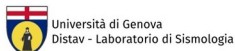
**Università di Genova**
Distav - Laboratorio di Sismologia

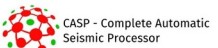
**CASP - Complete Automatic Seismic Processor**

MAGNITUDE (M_L): 5.8
Area: Trentino_SW_Lago_di_Garda_e_Lessini
Origin Time: 0000/00/00 00:00:00 (GMT +0)
Epicentre: 45.834 (°N) ; 11.066 (°E)
Depth: 9.0 (km)

EPICENTRE

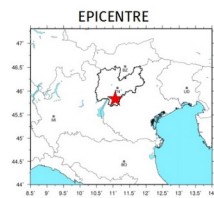

### Seismic shaking exposure

| Intensity (I_MCS): | ≤ III | IV | V | VI | VII | VIII | IX | X | ≥ XI | |
|---|---|---|---|---|---|---|---|---|---|---|
| Perceived Shaking: | Very light | Light | Moderate | Quite strong | Strong | Very strong | Severe | Very severe | Extreme | |
| Population[1]: | - | 70.7K | 107.3K | 187.4K | 72.5K | 71K | 0 | 0 | 0 | Total: 509.1K |
| Buildings of interest A[2]: | - | 198 | 375 | 231 | 89 | 161 | 0 | 0 | 0 | Total: 1.054 |
| Buildings of interest B[2]: | - | 270 | 390 | 618 | 235 | 201 | 0 | 0 | 0 | Total: 1.714 |

[1]ISTAT 2011 census estimation; [2]PAT 2022 census estimation.

SHAKING INTENSITY (interpolated values)

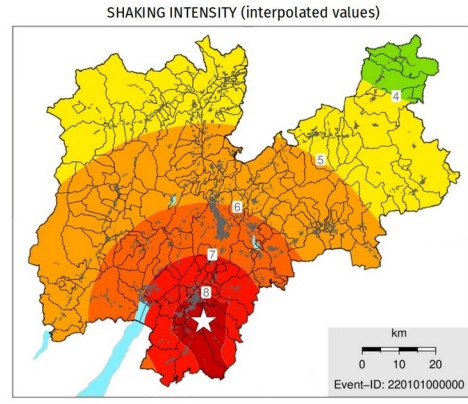

Event–ID: 220101000000

### MUNICIPALITIES EXPOSURE (first 20)

| I_MCS | Municipality | Population |
|---|---|---|
| VIII (8.5) | ALA | 8.792 |
| VIII (8.5) | ROVERETO | 39.954 |
| VIII (8.5) | TRAMBILENO | 1.468 |
| VIII (8.5) | VALLARSA | 1.364 |
| VIII (8.4) | MORI | 9.974 |
| VIII (8.3) | BRENTONICO | 4.021 |
| VIII (8.2) | ISERA | 2.754 |
| VIII (8.2) | TERRAGNOLO | 696 |
| VIII (8.0) | NOGAREDO | 2.075 |
| VIII (8.0) | VOLANO | 3.020 |
| VII (7.9) | VILLA LAGARINA | 3.825 |
| VII (7.9) | CALLIANO | 1.996 |
| VII (7.9) | FOLGARIA | 3.150 |
| VII (7.9) | POMAROLO | 2.418 |
| VII (7.9) | RONZO-CHIENIS | 987 |
| VII (7.8) | AVIO | 4.072 |
| VII (7.8) | NOMI | 1.312 |
| VII (7.7) | BESENELLO | 2.746 |
| VII (7.5) | ALDENO | 3.187 |
| VII (7.5) | ARCO | 17.798 |

SHAKING INTENSITY (actual values)

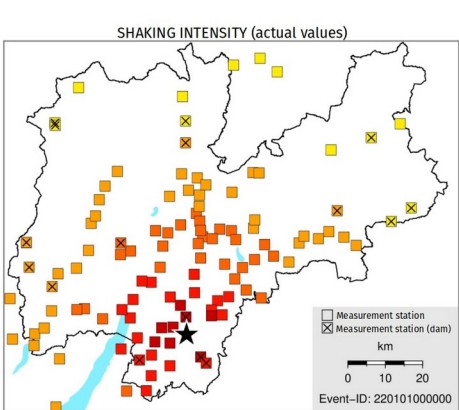

Measurement station
Measurement station (dam)
km
Event–ID: 220101000000

### DAMS (decreasing exposure)

| I_MCS | Accel. max (g) | Dams |
|---|---|---|
| VIII (8.5) | 0.3360 | San Colombano |
| VIII (8.3) | 0.3010 | Busa |
| VII (7.9) | 0.2470 | Speccheri |
| VII (7.3) | 0.1780 | Pra da Stua |
| VI (6.1) | 0.0800 | Ponte Pià |
| V (5.6) | 0.0580 | Murandin |
| V (5.3) | 0.0450 | Malga Boazzo |
| V (5.2) | 0.0390 | Malga Bissina |
| V (5.1) | 0.0380 | Mollaro |
| V (5.1) | 0.0350 | Costabrunella |
| IV (4.9) | 0.0310 | Santa Giustina |
| IV (4.7) | 0.0260 | Val Schener |
| IV (4.5) | 0.0220 | Pian Palù |
| IV (4.5) | 0.0210 | Malga Giumela |
| IV (4.4) | 0.0200 | Val Noana |
| IV (4.3) | 0.0180 | Forte Buso |

**Figure 10: Exposure map PDF for a strong earthquake simulated in Southern Trentino. See text (section 3) for description.**

**Appendix A**

List of low magnitude earthquakes recorded by at least one MEMS station, in the period July 2022–October 2023. The

event–MEMS distance is calculated considering the closest station to the hypocentre.

| ID | Date (yyyy-mm-dd) | UTC time (hh:mm:ss) | $M_L$ | Epicentral area (-) | Recording MEMS (#) | Event–MEMS distance (km) |
|----|-------------------|---------------------|-------|---------------------|--------------------|--------------------------|
| 1 | 2022-10-21 | 07:15:37 | 1.7 | Trentino | 2 | 14.0 |
| 2 | 2022-11-10 | 21:22:12 | 2.7 | Trentino | 2 | 46.7 |
| 3 | 2023-02-07 | 08:37:24 | 1.8 | Trentino | 1 | 16.3 |
| 4 | 2023-03-29 | 11:05:14 | 0.9 | Trentino | 1 | 18.0 |
| 5 | 2023-04-04 | 04:08:42 | 1.3 | Trentino | 1 | 10.7 |
| 6 | 2023-05-22 | 13:04:19 | 2.1 | Trentino | 1 | 44.4 |
| 7 | 2023-07-06 | 11:10:36 | 0.8 | Trentino | 1 | 4.7 |
| 8 | 2023-07-11 | 14:20:17 | 2.1 | Trentino | 4 | 5.0 |
| 9 | 2023-07-23 | 07:05:50 | 0.8 | Trentino | 1 | 3.1 |
| 10 | 2023-08-06 | 21:57:41 | 1.8 | Trentino | 2 | 10.6 |
| 11 | 2023-09-13 | 20:10:41 | 2.3 | Trentino | 6 | 5.2 |
| 12 | 2023-10-13 | 07:25:19 | 3.4 | Outside Trentino | 1 | 133.9 |
| 13 | 2023-10-25 | 13:45:37 | 4.2 | Outside Trentino | 13 | 79.7 |
| 14 | 2023-10-28 | 15:29:23 | 4.2 | Outside Trentino | 6 | 84.2 |

330