# Peer review of "A dense MEMS-based seismic network in populated areas: rapid es timation of exposure maps in Trentino (NE Italy)"

_Natural Hazards and Earth System Sciences, 2023_

## Referee Comment (RC1)

Comments to the paper: "A dense MEMS-based seismic network in populated areas: rapid estimation of exposure maps in Trentino (NE Italy)" by Scafidi et al.

This article presents the integration 76 low-cost accelerometer nodes into the existing permanent seismic network managed by the Autonomous Province of Trento. The purpose is to establish a denser seismic network that covers the whole Trentino area for a real-time monitoring and automatic generation of exposure maps. Indeed, the emphasis in the paper is the rapid estimation of exposure maps in Trentino.

The paper is certainly of interest, since at present it probably constitutes the first example in Italy of integration between a dense regional MEMS accelerometric network and a highly sensitive permanent network.

However, in its present form, the paper is lacking of some information and details which are important for the reader in order to evaluate the robustness of the results and conclusions. Therefore I suggest major revisions.

In general, I agree with the statements made in paragraph 4 "Summary and results" regarding the applicability of these very inexpensive MEMS for creating dense seismic monitoring networks also in urban environments. Although, I think that in this kind of networks can be useful deploying MEMS with different performance levels, where a percentage of the MEMS must have higher sensitivity and an ultra-low noise density ($< 1$ μg/ Hz).

English seems to be acceptable, but, because I am not a natural English speaker, I recommend to the Editorial Office to revise it.

Some specific comments:

I believe it is important to indicate in the paper when the network with 76 MEMS accelerometers began to work (fig. 1 reports 76 stations at 2023), as well as how many stations were present during the previous years. In the paper is reported only one example of earthquake referred to the 10 November 2022 event of ML 2.7.

**Therefore, I would like to know if the event reported was really the only earthquake that the MEMS network detected throughout 2022 and 2023.** As described in Cascone et al. (2021, *The Seismic Record. 1,20–26*) the ASX1000 has potential sensitivity to record local events with magnitude Mw > 2.5 in the 2–10 Hz frequency range. In the results they reported that the installed ASX1000 were able to detect nine small local earthquakes with 2.0 < ML < 3.0 between April 2020 and February 2021.

In Patanè et al. (2022, *Remote Sens., 14, 2583*) has been shown that the ADXL355 (about the ADXL355 see the next comments on paragraph 2.1) doesn't have a good signal-to-noise ratio for acceleration less than 1 cm/s$^2$. However, it is still possible to identify earthquakes with magnitudes less than 2.5 that happen less than 15 km away and determine the value of PGA.

Therefore, considering the density of accelerometric stations deployed in Trentino, **it would be useful to show some additional examples of earthquakes, if they are available, even if they have low magnitudes and recorded at few or at one station.**

Paragraph 2.1

In this paragraph, the authors provide information on the ASX1000, a low-cost MEMS sensor. It is said that the design and production of this sensor is owned by AD.EL. s.r.l. However, it may be more correct to state that Adel developed the board for housing and operating the MEMS accelerometer. This because the ADEL Srl does not manufacture MEMS sensors.

In the first instance, viewing the characteristics (noise floor of 25 µg/ Hz, sensitivity of 3.9 µg/LSB and bandwidth of 62.5 Hz at 250 Hz) reported in both the paper and the work of Cascone et al 2021, I supposed that the acceleremoter was a ADXL355 of the Analog Device. After that, I spoke with the ADEL and the technician sent me the information on the MEMS used in the ASX1000, which is, in fact, an ADXL355.

**Occur to consider, however, that the sensor not supports also ±1 g, as erroneously written in the paper.**

Furthermore, it is important to provide the necessary information regarding the ASX1000 board on which the MEMS is installed (e.g., if an SBC or MCU is present for data management, how the SeedLink protocol is implemented, how temporization is performed, etc.).

In figure 5, I note several problems. In particular, there is a shift in the noise floor curve of the ASX1000 and in the representative spectra responses of earthquakes for different moment magnitudes (dashed lines) measured at 10 km from the epicentre, with respect to the same curves reported in fig. 1 of Cascone et al. (2021), but also, for the representative response spectra, with those reported in fig. 2 of the paper of Nof et al. 2019 (*Earthquake Spectra, Volume 35, No. 1, pages 21–38*).

[Figure]

Fig. 1 – Cascone et al. 2021

[Figure]

Fig. 5 – Scafidi et al. 2023 present work

**I suggest that the authors carefully review Figure 5.**

Paragraph 2.2

In the paragraph 2.2 the authors report that the data are managed by a software package, called CASP (Complete Automatic Seismic Processor).

The authors should provide some details on the data transmission from the 76 installed MEMS stations to the central processing center that utilizes CASP. The delay associated with data transmission and the rate of data loss are examples of factors that can impact the efficiency and reliability of network and data availability itself. **I suggest that the authors include a figure showing a Gantt chart, in terms of working and not working, for the 76 accelerometric stations during their period of operation.**

Paragraph 2.3

Concerning the calculation of automatically exposure maps the authors used the empirical relationship of Faenza and Michelini (2010) to convert in intensity values.

**There is a new empirical relation for Italy by Oliveti, Faenza and Michelini (2022). Therefore, I think that this one should be considered instead of Faenza and MIchelini (2010).**

Paragraph 3.

In this paragraph the authors test the procedure of seismic shaking exposure considering a realistic emergency scenario for a moderate event, simulating an ML 5.8 earthquake in Southern Trentino (45.834 °N latitude, 11.066 °E longitude, 9.0 km depth), considering that it will represents a reference for the seismic potential of the Trentino region.

**How did the authors perform the numerical simulation? Which algorithm was used? Which source, path, and attenuation parameters were applied? It is important that the authors describe a comprehensive detail of the numerical simulations, since the above factors significantly influence the final results and the measured ground motion.**

In conclusion, my final recommendation is to perform a major revision. I think that the topic covered by the work is interesting and the authors could make it a lot better.

---

## Author Comment (AC1)

|   | Referee #1 | Reply by authors |
|---|---|---|
| | *General comments* | |
| 1 | This article presents the integration 76 low-cost accelerometer nodes into the existing permanent seismic network managed by the Autonomous Province of Trento. The purpose is to establish a denser seismic network that covers the whole Trentino area for a real-time monitoring and automatic generation of exposure maps. Indeed, the emphasis in the paper is the rapid estimation of exposure maps in Trentino.
The paper is certainly of interest, since at present it probably constitutes the first example in Italy of integration between a dense regional MEMS accelerometric network and a highly sensitive permanent network.
However, in its present form, the paper is lacking of some information and details which are important for the reader in order to evaluate the robustness of the results and conclusions. Therefore I suggest major revisions. | Dear Referee,
we would like to thank you for your comments and suggestions, and for the time you have spent evaluating our manuscript.
Answers to your specific comments are listed below.
In the revised text, we will add information and details, in order to better constraint results and conclusions. The use of the low-cost accelerometers and their integration with the Trentino permanent seismic network will be also clarified. |
| 2 | In general, I agree with the statements made in paragraph 4 "Summary and results" regarding the applicability of these very inexpensive MEMS for creating dense seismic monitoring networks also in urban environments. Although, I think that in this kind of networks can be useful deploying MEMS with different performance levels, where a percentage of the MEMS must have higher sensitivity and an ultra-low noise density ($< 1$ μg/ Hz). | At the moment all the MEMS stations are deployed using instruments with the same level of performance. However, the system infrastructure and the CASP software are able to integrate different instrumentation and to elaborate data from different sources (see also CASP description in Scafidi et al., 2018 SRL and Viganò et al., 2021 J. Seismol.). |
| 3 | English seems to be acceptable, but, because I am not a natural English speaker, I recommend to the Editorial Office to revise it. | The text will be carefully revised in order to improve English writing. |
| | *Specific comments* | |
| 4 | I believe it is important to indicate in the paper when the network with 76 MEMS accelerometers began to work (fig. 1 reports 76 stations at 2023), as well as how many stations were present during the previous years. | The starting date and the activity period of MEMS accelerometers will be more precisely stated. In addition, the definitive number of MEMS stations will be finally updated (73 instead of 76, at October 2023), both in the text and the figures. |
| 5 | In the paper is reported only one example of earthquake referred to the 10 November 2022 event of ML 2.7.
Therefore, I would like to know if the event reported was really the only earthquake that | The earthquakes presented in the manuscript (November $10^{th}$ 2022, $M_L$ 2.7, Fig. 7; July $11^{th}$ 2023, $M_L$ 2.1, Fig. 9) are the two better recorded in the considered period (July 2022-October 2023; updated). |

| | | |
|---|---|---|
| | the MEMS network detected throughout 2022 and 2023.
 As described in Cascone et al. (2021, The Seismic Record. 1,20–26) the ASX1000 has potential sensitivity to record local events with magnitude Mw > 2.5 in the 2–10 Hz frequency range. In the results they reported that the installed ASX1000 were able to detect nine small local earthquakes with 2.0 < ML < 3.0 between April 2020 and February 2021.
 In Patanè et al. (2022, Remote Sens., 14, 2583) has been shown that the ADXL355 (about the ADXL355 see the next comments on paragraph 2.1) doesn't have a good signal-to-noise ratio for acceleration less than 1 cm/s$^2$. However, it is still possible to identify earthquakes with magnitudes less than 2.5 that happen less than 15 km away and determine the value of PGA.
 Therefore, considering the density of accelerometric stations deployed in Trentino, it would be useful to show some additional examples of earthquakes, if they are available, even if they have low magnitudes and recorded at few or at one station. | Additional examples will be supplied. In particular, a complete list of seismic events which were recorded by at least one MEMS station will be added as Appendix to the manuscript. This further information will be commented in the revised text. Even if the MEMS sensors are principally aimed to register the strong ground motion, the possibility to use low-cost accelerometers to register low magnitude earthquakes at small hypocentral distances will be also discussed. Moreover, in the figure attached here below a few examples about the quality of very low magnitude recordings are shown. |
| 6 | Paragraph 2.1
 In this paragraph, the authors provide information on the ASX1000, a low-cost MEMS sensor. It is said that the design and production of this sensor is owned by AD.EL. s.r.l. However, it may be more correct to state that Adel developed the board for housing and operating the MEMS accelerometer. This because the ADEL Srl does not manufacture MEMS sensors. | We agree that AD.EL srl is not the manufacturer of the MEMS sensor. For this reason, the sentence will be accordingly rephrased. |
| 7 | In the first instance, viewing the characteristics (noise floor of 25 μg/ Hz, sensitivity of 3.9 μg/LSB and bandwidth of 62.5 Hz at 250 Hz) reported in both the paper and the work of Cascone et al 2021, I supposed that the acceleremoter was a ADXL355 of the Analog Device. After that, I spoke with the ADEL and the technician sent me the information on the MEMS used in the ASX1000, which is, in fact, an ADXL355.
 Occur to consider, however, that the sensor not supports also ±1 g, as erroneously written in the paper. | The sentence will be modified, as suggested. |
| 8 | Furthermore, it is important to provide the | The MEMS instrument considered in this |

| | | |
|---|---|---|
| | necessary information regarding the ASX1000 board on which the MEMS is installed (e.g., if an SBC or MCU is present for data management, how the SeedLink protocol is implemented, how temporization is performed, etc.). | study is a new version of the ASX1000 presented in Cascone et al. (2021). In fact, even if both mount the ADXL355 accelerometer, some improvements were made (e.g., low-pass digital filters, 4G LTE modem, low-power with battery management). To better clarify this point, the MEMS instrument will be renamed as ASX1000v2 (upgraded version of the original ASX1000) in the revised text and figures. Further information about the ASX1000v2 board will be added. In particular, about data management (MCU model), the SeedLink protocol, and temporization (performed using the NTP protocol). |
| 9 | In figure 5, I note several problems. In particular, there is a shift in the noise floor curve of the ASX1000 and in the representative spectra responses of earthquakes for different moment magnitudes (dashed lines) measured at 10 km from the epicentre, with respect to the same curves reported in fig. 1 of Cascone et al. (2021), but also, for the representative response spectra, with those reported in fig. 2 of the paper of Nof et al. 2019 (Earthquake Spectra, Volume 35, No. 1, pages 21–38). I suggest that the authors carefully review Figure 5. | Figure 5 will be carefully checked and graphically corrected. In particular, the noise floor curve of the accelerometric sensor and the representative response spectra will be modified, also according to Cascone et al. (2021) and Nof et al. (2019). |
| 10 | Paragraph 2.2 In the paragraph 2.2 the authors report that the data are managed by a software package, called CASP (Complete Automatic Seismic Processor). The authors should provide some details on the data transmission from the 76 installed MEMS stations to the central processing center that utilizes CASP. The delay associated with data transmission and the rate of data loss are examples of factors that can impact the efficiency and reliability of network and data availability itself. I suggest that the authors include a figure showing a Gantt chart, in terms of working and not working, for the 76 accelerometric stations during their period of operation. | In order to verify the activity of the monitoring system, the CASP software is set to automatically check data transmission and data availability from the fluxes of all the network stations (both highly sensitive sensors and low-cost accelerometers) (see description in Viganò et al., 2021, J. Seismol.). In particular, each hour an automatic control is performed, to monitor data availability and time latency. If any problem arises, SeedLink is rebooted and technical alerts are sent to seismologists. For this reason, the results from each automatic check are not stored and not made available later in time. In addition, even if the continuous data flux of the highly sensitive sensors is stored, that from the low-cost accelerometers is not. In fact, automatic storing is performed by CASP only for the portions of the MEMS seismic traces corresponding to an earthquake detected by |

| | | the system. |
|---|---|---|
| | | However, data latency and data loss can be evaluated for selected periods (e.g., analyzing a one-week distribution of data). In this case, the typical average latency is in the order of about 15 s, while the data flux of all the MEMS stations is continuous and complete at about 99.5 %. This information, together with a comprehensive description on data transmission will be stated in the revised text. |
| 11 | Paragraph 2.3 Concerning the calculation of automatically exposure maps the authors used the empirical relationship of Faenza and Michelini (2010) to convert in intensity values. There is a new empirical relation for Italy by Oliveti, Faenza and Michelini (2022). Therefore, I think that this one should be considered instead of Faenza and Michelini (2010). | As suggested, the empirical relation for Italy by Faenza and Michelini (2010) will be substituted by the more recent one by Oliveti, Faenza and Michelini (2022). Figures 8, 9 and 10 will be consequently modified. |
| 12 | Paragraph 3 In this paragraph the authors test the procedure of seismic shaking exposure considering a realistic emergency scenario for a moderate event, simulating an ML 5.8 earthquake in Southern Trentino (45.834 °N latitude, 11.066 °E longitude, 9.0 km depth), considering that it will represents a reference for the seismic potential of the Trentino region. How did the authors perform the numerical simulation? Which algorithm was used? Which source, path, and attenuation parameters were applied? It is important that the authors describe a comprehensive detail of the numerical simulations, since the above factors significantly influence the final results and the measured ground motion. | The emergency scenario of Fig. 10 does not represent a complete numerical simulation, but a "simplified simulation" with the aim of evaluating a realistic scenario in the case of a strong earthquake in Southern Trentino (compare also with point #3 of Referee #2). This point will be more clearly stated in the revised text, also specifying the method used (magnitude assignment and seismic attenuation calculation at each station of the network). |
| 13 | In conclusion, my final recommendation is to perform a major revision. I think that the topic covered by the work is interesting and the authors could make it a lot better. | The manuscript will be carefully checked, in order to generally improve comprehension and readability. |

Figure (see point #5)
$M_L$, local magnitude; HypoDist, hypocentral distance

---

## Author Comment (AC2)

| | Referee #2 | Reply by authors |
|---|---|---|
| | *General comments* | |
| 1 | I have reviewed the manuscript nhess-2023-143 'A dense MEMS-based seismic network in populated areas: rapid estimation of exposure maps in Trentino (NE Italy)' by Scafidi et al. The work presents a novel accelerometric network made up 76 stations with MESM in the Trentino region in Italy. The purpose of the network is to complement the existing accelerometric network in the area, increasing the density of stations in inhabited areas of the region. The novel MEMS-based network can allow improving the rapid response phases after a moderate or a large earthquake by providing a more accurate measure of ground motion intensity with respect to estimates derived by the standard network and the use of ground motion prediction equations. The work is well organized, and the topic is suitable for NHESS. | Dear Referee, we would like to thank you for your concerns and suggestions, and for the time you have spent evaluating our manuscript. Answers to your specific comments are listed below. |
| | **Specific comments** | |
| 2 | I have only a few minor concerns/suggestions for the Authors. I agree that MEMS can be perfect for the purpose of the work (the strong ground motion). However, throughout the paper it is mentioned a few times that the MEMS would fine to record weaker seismicity too. The sentence is very general and could be misleading for readers. Indeed, the possibility to record the earthquake signals depends on the hypocentral distance. It is true that in Fig. 5 the MEMS allows to record a bit of frequencies also for a Mw 2.5 earthquakes, but this is at 10 km of distances only. I imagine that if the hypocentral distance increases the MEMS it is too noisy to record the source signals (Fig.7 nicely show this effect). Moreover, the amplitudes of sources in Fig. 5 are also strongly influenced by the attenuation. In conclusion, I suggest clarifying the effective utility of MEMS with respect to weaker seismicity. | We agree that MEMS stations are not useful to register low magnitude earthquakes ($M_L$ <3) at relatively large hypocentral distances (> 10 km). In the revised manuscript, we will better explain this point, also more clearly stating the primary aim of the MEMS network presented in this study (compare also with point #5 of Referee #1). That is, to perform densely distributed and quasi real-time strong motion data and exposure maps in the urbanized areas of Trentino, based on locations performed by using the permanent seismic network of the Autonomous Province of Trento. In fact, in the case of strong earthquakes, MEMS stations can be crucial both in order to register strong motion data and to integrate the automatic location system (for example, with additional phase arrivals in the epicentral area). |
| 3 | Did you consider directivity and finite fault effects in the worst-case scenario of Fig. 10? While these aspects do not limit the advantage of having a MEMS network in urban area, the scenario in terms of losses | We did not consider directivity and finite fault effects in the scenario shown in Fig. 10. For this reason, we agree that the scenario could be even worse than presented. This point will be accordingly clarified in the |

| | | |
|---|---|---|
| | could be even worst of the one shown. Maybe, it is sufficient clarify that this is a simplified simulation. | revised text, specifying that is a simplified simulation. |
| 4 | Page 1. Line 24. "even more" it is not clear more with respect to what. | The sentence will be rephrased, as suggested. |
| 5 | Page 4. Line 122. Clarify the comparison with instrumented dams in Fig.6. I agree that is an important and useful information, but clarify the sentence | The sentence will be clarified, in order to highlight that intensity values at 16 instrumented dams (definitive number updated on Fig. 6) are also available on the exposure map PDF, in the case of strong earthquakes. |
| 6 | In conclusion, I suggest minor revision | According to the received suggestions, the manuscript will be improved. |

---

## Author Response (AR2)

| | Referee #3 | Reply by authors |
|---|---|---|
| | *General comments* | |
| 1 | This paper presents the MEMS-based seismic network whose records are used for a rapid estimation of moderate to strong earthquake exposure maps in Trentino (NE Italy), an area at risk of earthquakes. The results are in pdf documents, which are useful in disaster management efforts in the case of a strong earthquake occurence. The paper is certainly of interest and I strongly recommend its publication. Since important revision suggestions were made by the former reviewers, and the authors have made them, I consider the paper ready for publication only after some minor revisions to a few figure captions. | Dear Referee, We would like to thank you for your suggestions. Answers to your specific comments are listed below. |
| | **Specific comments** | |
| 2 | Fig. 1 - also explain the red triangles. | The red triangles are already explained in the graphical legend. |
| 3 | Fig. 5 - bracketed black line and corresponding red line as - Noise floor of the ASX1000v2 MEMS (black line) compared to typical ground motion amplitudes of earthquakes measured at 10 km from the epicentre for different moment magnitudes (dashed lines). The new high noise model (NHNM-red line) from Peterson (1993) is also shown for reference. | The caption has been accordingly modified. |
| 4 | Fig. 6 - the colour of the building symbol would be changed so as not to be confused with the symbol for population density. Maybe green? | The colour of the building symbol has been changed into red, and dams into yellow for better readability. |
| 5 | Fig. 7 - specify red and blue vertical lines and note that the sensors are with blue triangles. | Red and blue vertical lines are already explained in the caption. A sentence on the sensor location has been also added. |
| 6 | Fig. 8 - you must specify T=corner period and Sa=spectral acceleration. For the other parameters, indicate at least the page where they can be found - pg. 5. | The explanations for "T" and "Sa" have been improved in the main text. Regarding the other parameters, the reference of "section 3" has been specified. |
| 7 | Fig. 9,10 - specify page(s) where explanations are. | The reference of "section 3" has been specified in both the captions. |

---

## Author Response (AR3)

As mentioned in our earlier email, concerning Figure 8, we would kindly prefer to label it as a "screenshot" (without converting it into an actual table within the manuscript) to emphasize its origin from the dedicated web portal. Thank you for your understanding.